# Comparative outcomes of DSAEK and DMEK in eyes with prior Ahmed glaucoma valve implantation

Do-Hyeon An[1], Gyeongmin Yoo[1], Ji-Yun Song[1], Jin-Uk Beak[2], So-Hyang Chung[1], Hyun-Seung Kim[1], Yong-Soo Byun[1]*

**1** Department of Ophthalmology, Seoul St. Mary's Hospital, College of Medicine, The Catholic University of Korea, Seoul, Korea, **2** Department of Ophthalmology, Incheon St. Mary's Hospital, College of Medicine, The Catholic University of Korea, Incheon, Korea

* mdbyun@catholic.ac.kr

## Abstract

### Background

To compare the outcomes of Descemet stripping automated endothelial keratoplasty (DSAEK) and Descemet membrane endothelial keratoplasty (DMEK) in eyes with prior Ahmed glaucoma valve (AGV) implantation.

### Methods

This retrospective study included 27 eyes from 27 patients who underwent endothelial keratoplasty (16 DSAEK, 11 DMEK) between 01/01/2020 and 31/12/2022. All eyes had a prior AGV implantation and a minimum follow-up of 12 months. Best-corrected visual acuity (BCVA), endothelial cell density (ECD), central corneal thickness (CCT), complications, and graft survival were compared between groups.

### Results

The donor age was significantly younger in the DSAEK group (52.94 ± 11.17 years vs. 62.36 ± 10.02 years, P = 0.034). The DSAEK group had significantly higher ECD at 6 and 12 months (1583.2 ± 562.6 and 1441.1 ± 502.6 cells/mm²) compared to the DMEK group (1109.5 ± 484.8 and 1046.6 ± 327.3 cells/mm², P = 0.040 and P = 0.043). However, this difference was no longer significant after covariate analysis with donor ECD. Postoperative BCVA improved significantly in both groups, with no significant differences between them. Complication rates were comparable. The 1-year survival rates were 87% for DSAEK and 91% for DMEK, the 2-year survival rates were 78% for DSAEK and 74% for DMEK. Mean graft survival was 28.7 ± 3.3 months for DSAEK and 24.7 ± 2.8 months for DMEK (P = 0.777).

**Data availability statement:** Data cannot be shared publicly because of ethical and privacy restrictions. This study adhered to the principles of the Declaration of Helsinki and was approved by the Institutional Review Board of Seoul St. Mary's Hospital, College of Medicine, The Catholic University of Korea (IRB number: KC22RISI0525/2022-1648-0004). The requirement for informed consent was waived due to the retrospective nature of the study and the use of fully anonymized data. All data were fully anonymized prior to analysis, and the investigators did not have access to any identifying information. Data are available from the Institutional Review Board of Seoul St. Mary's Hospital (contact via irb@catholic.ac.kr) for researchers who meet the criteria for access to confidential data.

**Funding:** This research was partially supported by Basic Science Research Program through the National Research Foundation of Korea(NRF) funded by the Ministry of Education (2022R1C1C1011531).

**Competing interests:** The authors have declared that no competing interests exist.

## Conclusions

DSAEK and DMEK demonstrated comparable visual outcomes, complication rates, and graft survival in eyes with prior AGV implantation. While both procedures appear to be viable options for this complex patient population, these findings should be confirmed in larger studies.

## Introduction

Corneal endothelial cell loss and subsequent decompensation are significant complications following glaucoma surgeries, particularly those involving glaucoma drainage devices (GDDs). One study reported a 12.3% decrease in endothelial cell density (ECD) 12 months postoperatively in eyes with Ahmed glaucoma valve (AGV) implants, compared to 3.2% in eyes that underwent trabeculectomy during the same period [1]. Approximately 5% of eyes undergoing GDD implantation develop corneal decompensation, with 2.9% requiring keratoplasty [2].

Endothelial keratoplasty (EK) is the preferred treatment option for corneal endothelial decompensation, offering faster visual recovery and lower rejection rates than penetrating keratoplasty [3,4]. Among EK techniques, Descemet membrane endothelial keratoplasty (DMEK) provides better visual outcomes and lower rejection rates than Descemet stripping automated endothelial keratoplasty (DSAEK) due to its greater anatomical restoration [5–7]. Whether these advantages persist in eyes with GDDs remains unclear, as DSAEK offers greater surgical ease and reproducibility due to the use of thicker donor grafts. Several studies have examined DSAEK or DMEK outcomes in eyes with prior glaucoma surgery, but most included a heterogeneous mix of filtering procedures and different drainage devices [8–12].

Despite growing experience with both techniques, direct comparisons between DSAEK and DMEK in this population remain limited. This study aimed to address this knowledge gap by comparing the outcomes of DSAEK and DMEK in eyes with a single type of GDDs, namely, AGVs. The analysis focused on visual acuity, ECD, central corneal thickness (CCT), postoperative complications, and graft survival to provide evidence-based guidance for surgical decision-making in this challenging patient population.

## Materials and methods

This retrospective study included patients who underwent endothelial keratoplasty between 01/01/2020 and 31/12/2022, with postoperative evaluations performed at regular follow-up visits during the 12-month postoperative period. The inclusion criteria consisted of the presence of an AGV in the operated eye, no history of prior corneal transplantation, and a minimum follow-up period of 12 months. All eligible cases were consecutively included, with no exclusions among eyes with prior AGV implantation undergoing DSAEK or DMEK. This study adhered to the principles of the Declaration of Helsinki and was approved by the Institutional Review Board (IRB number: KC22RISI0525/2022-1648-0004). The requirement for informed consent

was waived due to the retrospective nature of the study and the use of fully anonymized data. All data were fully anonymized prior to analysis, and the investigators did not have access to any identifying information.

Medical records were reviewed for glaucoma diagnosis, glaucoma surgeries, slit-lamp findings, best-corrected visual acuity (BCVA), intraocular pressure (IOP), surgical history, ocular and systemic comorbidities, ECD, CCT, and postoperative complications, including rebubbling, IOP elevation, and cystoid macular edema (CME). Glaucoma diagnosis was based on prior clinical diagnosis by glaucoma specialists and included primary open-angle glaucoma, secondary glaucoma, uveitic glaucoma, pseudoexfoliative glaucoma, neovascular glaucoma, and iridocorneal endothelial syndrome. Eyes with a single AGV, multiple AGVs (≥2), and those with prior trabeculectomy or microinvasive glaucoma surgery in addition to AGV implantation were included. Lens status was recorded as phakic, intraocular lens in the bag, or scleral-fixated intraocular lens.

Postoperative BCVA, IOP, ECD, and CCT were assessed at every visit throughout the 12-month postoperative period. BCVA was measured using a Snellen chart and converted to logMAR units. IOP was measured using a non-contact pneumotonometer (CT-80A; Canon Inc., Tokyo, Japan). ECD and CCT were assessed using specular microscopy (CellChek XL; Konan Medical, Nishinomiya, Japan) and an ultrasound pachymeter (SP-3000; Tomey Corporation, Nagoya, Japan), respectively. Axial length (AL) and anterior chamber depth (ACD) were measured preoperatively using optical biometry (IOLMaster 700; Carl Zeiss Meditec, Jena, Germany). Graft attachment was evaluated postoperatively using slit-lamp biomicroscopy and anterior segment optical coherence tomography (ANTERION; Heidelberg Engineering, Heidelberg, Germany). Retinal and macular status were evaluated at postoperative 1 and 3 using ultra-widefield imaging (Optos Silverstone; Optos PLC, Dunfermline, UK) and spectral-domain OCT (Heidelberg Retina Angiograph OCT; Heidelberg Engineering, Heidelberg, Germany).

Donor corneal characteristics, including age, preservation-to-operation time, ECD, and graft thickness and size, were also analyzed. All donor corneal tissues were preserved in Optisol-GS and transported at 4°C according to standard eye bank protocols. Postoperative endothelial cell loss (ECL) was additionally calculated and expressed as a percentage (%ECL) relative to the donor ECD using the following formula: %ECL = [(donor ECD − postoperative ECD)/ donor ECD] × 100.

Primary graft failure was defined as persistent corneal edema without improvement within the first three months postoperatively. Secondary graft failure was defined as the reemergence of corneal edema after initial postoperative improvement, occurring at any time during the 12-month follow-up period. Postoperative IOP elevation was defined as an IOP exceeding 21 mm Hg and/or the need for additional glaucoma medications at any time during the postoperative follow-up period. Data extraction and analysis were performed by independent observers who were not involved in surgical procedures or clinical care to minimize selection bias.

The choice between DSAEK and DMEK in eyes with AGVs was based on the surgeon's preference. DMEK was performed by a single surgeon (Y-.S.B.), whereas DSAEK was performed by two surgeons (H-.S.K. and S-.H.C.) using standardized techniques. All phakic eyes were left phakic at the time of DMEK or DSAEK, and no simultaneous or subsequent cataract surgery was performed during the study period. For DSAEK, precut donor tissue with a graft thickness less than 100 μm was obtained from a nonprofit eye bank. After removing the corneal epithelium to improve visualization, a circular mark 0.5 mm larger than the graft diameter was made on the corneal surface. Following viscoelastic (Hyalu®; Hanmi Pharm. Co., Ltd., Seoul, Korea) injection, Descemet's membrane was scored along the circular mark and carefully stripped using a modified reverse Sinskey hook (Moria SA, Antony, France). Inferior iridectomy was performed, and the viscoelastic material was thoroughly removed using irrigation-aspiration. The donor cornea was trephined and loaded into either an Endoglide (Coronet®, Network Medical Products, North Yorkshire, UK) or Macaluso's inserter (J3760.2, e.janach, Italy). The graft was inserted through a 3.5- to 4.5-mm incision using the pull-through technique, positioned centrally, and secured with filtered room air. Patients were maintained in the supine position for at least four hours postoperatively.

 

For DMEK, precut and preloaded grafts in bent-pipette glass tubes were obtained from a nonprofit eye bank. After Descemet's membrane removal and viscoelastic clearance, the graft was inserted through a 2.4- to 2.7-mm incision. The scrolled graft was unfolded by tapping the corneal surface with a cannula. Filtered room air was used for graft attachment, followed by 2–4 hours of supine positioning. Patients were strictly instructed to remain in the supine position and avoid eye rubbing after surgery.

Rebubbling with filtered room air or 20% sulfur hexafluoride was performed when graft detachment involved at least one-third of the graft area postoperatively. When the AGV tube was excessively long on intraoperative assessment, valve tip trimming was performed before graft insertion to avoid interference with graft positioning.

Statistical analysis was conducted using SPSS software (version 21.0; SPSS Inc., Chicago, IL, USA). After assessing data normality, mean values were compared using Mann-Whitney U test, the Wilcoxon signed-rank test, and frequencies were analyzed using the Fisher exact test. Covariate analysis (Analysis of Covariance) was used to compare postoperative ECD to account for donor ECD. Kaplan-Meier curves were used to calculate survival rates, and groups were then compared using the log-rank test.

## Results

Twenty-seven eyes from 27 patients were included (16 underwent DSAEK and 11 underwent DMEK) at baseline. At each postoperative time point, eyes that had experienced graft failure by that visit were excluded from the analysis. Consequently, the number of eyes analyzed varied across time points: the DSAEK and DMEK groups included 16 and 11 eyes at 1 week, 1 month, and 3 months; 16 and 10 eyes at 6 months; and 14 and 10 eyes at 12 months, respectively. Table 1 summarizes the characteristics of the study groups and the donor corneas. Both groups showed no difference in the diagnosis of glaucoma, the variations of AGV surgery, and time from AGV to EK surgery. All implanted AGV tubes were placed in the anterior chamber. AL was significantly longer between the DSAEK (24.95 ± 1.48 mm) and the DMEK group (23.87 ± 1.27 mm) (P = 0.039). ACD, lens status, and history of prior vitrectomy were not different between two groups. The donor age was significantly younger in the DSAEK group (52.94 ± 11.17 years vs. 62.36 ± 10.02 years, P = 0.034), whereas the donor ECD was not different between two groups. The mean graft thickness of DSAEK group was 58.14 ± 11.88 μm (range: 45–88 μm), which corresponded to the ultra-thin DSAEK.

Postoperative visual acuity significantly improved compared with preoperative values in both groups at 1, 3, 6, and 12 months (all P < 0.05), with no significant differences between the groups at any time point (Table 2).

While ECD showed no significant differences between the groups at 1 and 3 months postoperatively, the DSAEK group (1583.2 ± 562.6 and 1441.1 ± 502.6 cells/mm²) demonstrated significantly higher ECD compared to the DMEK group (1109.5 ± 484.8 and 1046.6 ± 327.3 cells/mm², P = 0.040 and P = 0.043, respectively) at 6 and 12 months (Table 3). However, when covariate analysis was performed with donor ECD as a covariate, no significant differences in ECD were observed between the two groups at any time point. Postoperative %ECL showed no statistically significant differences between the DSAEK and DMEK groups at 1, 3, 6, or 12 months postoperatively (Table 4).

CCT decreased significantly from preoperative values beginning at one month postoperatively in both groups (Table 5). Despite the inherent thickness of DSAEK grafts (58.14 ± 11.88 μm), postoperative CCT showed no significant differences between the two groups throughout the observation period. Complication rates, including rebubbling, IOP requiring additional medications, and CME, showed no significant differences between the groups (Table 6). In two DMEK cases, valve tip trimming was required during surgery, because the valve tip could potentially interfere with DMEK graft unfolding procedure. One of these patients subsequently underwent valve tip repositioning into the sulcus at 4 months postoperatively due to uncontrolled IOP elevation and iris chafing. All other episodes of postoperative IOP elevation were successfully controlled with additional glaucoma medications. Nevertheless, graft survival was favorable, with a ECD > 1000 cells/mm² at postoperative month 12, comparable to the overall cohort. In the other patient, there was no tip malposition that required the surgical revision perioperatively.

 

**Table 1. Patient and donor characteristics.**

| | DSAEK (n = 16) | DMEK (n = 11) | Mean Difference (95% CI) | P value |
|---|---|---|---|---|
| | Mean ± Standard deviation or N (%) | | | |
| **Recipient factors** | | | | |
| Age | 71.06 ± 9.04 | 63.27 ± 13.31 | 7.79 (−1.04 to 16.62) | .138* |
| Sex (M:F) | 14:2 | 9:2 | | |
| Diagnosis of glaucoma | | | | |
| POAG | 1 (6.2%) | 2 (18.2%) | | .545† |
| Secondary glaucoma | 7 (43.8%) | 4 (36.4%) | | |
| Uveitic glaucoma | 5 (31.2%) | 1 (9.1%) | | |
| PEX | 2 (12.5%) | 2 (18.2%) | | |
| NVG | 0 (0.0%) | 1 (9.1%) | | |
| ICE syndrome | 1 (6.2%) | 1 (9.1%) | | |
| Glaucoma surgery | | | | |
| AGV = 1 | 13 (81.2%) | 8 (72.7%) | | .468† |
| AGV and prior trabeculectomy or MIGS) | 0 (0.0%) | 1 (9.1%) | | |
| AGV ≥ 2 | 3 (18.8%) | 2 (18.2%) | | |
| Time from AGV to EK (years) | 6.31 ± 7.06 | 6.16 ± 6.82 | 0.15 (−5.16 to 5.46) | .844* |
| Axial length (mm) | 24.95 ± 1.48 | 23.87 ± 1.27 | 0.18 (−0.39 to 0.75) | .039* |
| Anterior chamber depth (mm) | 4.52 ± 0.89 | 4.34 ± 0.63 | 1.08 (0.04 to 2.12) | .526* |
| Lens status | | | | |
| Phakic | 1 (6.2%) | 2 (18.2%) | | .703† |
| IOL in the bag | 11 (68.8%) | 7 (63.6%) | | |
| Scleral fixated IOL | 4 (25.0%) | 2 (18.2%) | | |
| Vitrectomized eye | 2(25%) | 1(18%) | | .782† |
| Underlying disease | | | | |
| Diabetes mellitus | 3 (18.8%) | 2 (18.2%) | | 1.000† |
| Hypertension | 6 (37.5%) | 3 (27.3%) | | |
| Autoimmune | 1 (6.3%) | 0 (0.0%) | | |
| **Donor factors** | | | | |
| Donor age | 52.94 ± 11.17 | 62.36 ± 10.02 | −9.42 (−18.07 to −0.77) | .034* |
| ECD (cells/mm²) | 2874.6 ± 200.6 | 2715.3 ± 272.6 | 159.3 (−27.92 to 346.52) | .103* |
| Preservation to transplant time (hours) | 128.94 ± 25.65 | 125.82 ± 32.07 | 3.12 (−19.78 to 26.02) | .964* |
| Graft thickness (μm) | 58.14 ± 11.88 | − | | |
| Graft diameter (mm) | 7.95 ± 0.16 | 8.02 ± 0.18 | −0.07 (−0.21 to 0.07) | .455* |

All values are presented as mean ± standard deviation or percentage (%).

DSAEK, Descemet stripping automated endothelial keratoplasty; DMEK, Descemet's membrane endothelial keratoplasty; AGV, Ahmed glaucoma valve; EK, endothelial keratoplasty; POAG, primary open-angle glaucoma; PEX, pseudoexfoliative glaucoma; NVG, neovascular glaucoma; ICE, iridocorneal endothelium; IOL: Intraocular lens; IOP, intraocular pressure; ECD: Endothelial cell density; MIGS: Microinvasive glaucoma surgery.

* Mann-Whitney test; † Fisher exact test.

All graft failures observed in our study were secondary failures, occurring in 5 of 16 eyes in the DSAEK group and 3 of 11 eyes in the DMEK group. Kaplan-Meier analysis estimated mean survival times of 28.7 ± 3.3 months for DSAEK and 24.7 ± 2.8 months for DMEK, with no significant difference between the groups (P = 0.777, Fig 1). The 1-year survival rates were found to be 87% for DSAEK and 91% for DMEK, while the 2-year survival rates were 78% for DSAEK and 74% for DMEK.

**Table 2. Comparison of visual acuity between DSAEK and DMEK.**

| logMAR VA | DSAEK (n = 16) | DMEK (n = 11) | Mean Difference | P value |
|---|---|---|---|---|
| | Mean ± Standard deviation | | (95% CI) | |
| Preoperative | 1.95 ± 0.48 | 2.15 ± 0.52 | −0.20 (−0.59 to 0.19) | |
| 1 week | 1.26 ± 0.68 | 1.33 ± 0.78 | −0.07 (−0.65 to 0.51) | .803 |
| 1 month | 0.93 ± 0.66* | 0.72 ± 0.54* | 0.21 (−0.29 to 0.71) | .387 |
| 3 months | 0.94 ± 0.73* | 0.81 ± 0.61* | 0.13 (−0.42 to 0.68) | .634 |
| 6 months‡ | 0.89 ± 0.66* | 0.84 ± 0.61* | 0.05 (−0.47 to 0.57) | .823 |
| 12 months§ | 0.98 ± 0.76* | 1.08 ± 0.74* | −0.10 (−0.71 to 0.51) | .733 |

VA, visual acuity; DSAEK, Descemet stripping automated endothelial keratoplasty; DMEK, Descemet's membrane endothelial keratoplasty.

* Significant difference with the preoperative value; ‡ 6-month analysis: DSAEK n = 16, DMEK n = 10; § 12-month analysis: DSAEK n = 14, DMEK n = 10.

**Table 3. Comparison of endothelial cell density between DSAEK and DMEK.**

| ECD (cells/mm²) | DSAEK (n = 16) | DMEK (n = 11) | Mean Difference | P value | |
|---|---|---|---|---|---|
| | Mean ± Standard deviation | | (95% CI) | | |
| Donor | 2874.6 ± 200.6 | 2715.3 ± 272.6 | 159.30 (−29.42 to 348.02) | .092* | .120† |
| 1 month | 1685.3 ± 576.7 | 1550.9 ± 748.7 | 134.40 (−390.59 to 659.39) | .647* | .668† |
| 3 months | 1704.5 ± 611.5 | 1486.7 ± 578.7 | 217.80 (−236.88 to 672.48) | .393* | .410† |
| 6 months‡ | 1583.2 ± 562.6 | 1109.5 ± 484.8 | 473.70 (76.11 to 871.29) | .040* | .136† |
| 12 months§ | 1441.1 ± 502.6 | 1046.6 ± 327.3 | 394.50 (81.35 to 707.65) | .043* | .149† |

ECD, endothelial cell density; DSAEK, Descemet stripping automated endothelial keratoplasty; DMEK, Descemet's membrane endothelial keratoplasty.

* Mann-Whitney test; † Analysis of Covariance (ANCOVA) and Repeated Measures ANCOVA; ‡ 6-month analysis: DSAEK n = 16, DMEK n = 10;
§ 12-month analysis: DSAEK n = 14, DMEK n = 10.

**Table 4. Comparison of endothelial cell loss (%) between DSAEK and DMEK.**

| ECL (%) | DSAEK (n = 16) | DMEK (n = 11) | P value* |
|---|---|---|---|
| | Mean ± Standard deviation | | |
| 1 month | 40.8 ± 19.8 | 44.1 ± 28.2 | .752 |
| 3 months | 40.5 ± 21.8 | 46.0 ± 21.0 | .548 |
| 6 months‡ | 44.8 ± 20.3 | 59.2 ± 17.0 | .076 |
| 12 months§ | 49.9 ± 16.8 | 60.9 ± 13.8 | .129 |

ECL, endothelial cell loss; DSAEK, Descemet stripping automated endothelial keratoplasty; DMEK, Descemet's membrane endothelial keratoplasty. * Mann-Whitney test; ‡ 6-month analysis: DSAEK n = 16, DMEK n = 10;
§ 12-month analysis: DSAEK n = 14, DMEK n = 10.

## Discussion

This study directly compared DSAEK and DMEK outcomes in eyes with prior AGV implantation. While previous studies examining EK after glaucoma surgery included a heterogeneous mix of procedures and drainage devices [8–12], we specifically focused on eyes with a single type of GDD, namely, AGV, allowing controlled comparison between DSAEK and DMEK. Although our cohort included various underlying glaucoma diagnoses (POAG 40.7%, secondary glaucoma 22.2%, uveitic glaucoma 14.8%, and others), the distribution was comparable between DSAEK and DMEK groups (P = 0.545), and all eyes exhibited endothelial decompensation, likely due to the AGV. Our findings suggest comparable visual outcomes, complication rates, and graft survival between the two techniques in this challenging population, though these results should be interpreted cautiously given our limited sample size.

**Table 5. Comparison of central corneal thickness between DSAEK and DMEK.**

| CCT (μm) | DSAEK (n = 16) | DMEK (n = 11) | Mean Difference | P value |
|---|---|---|---|---|
| | Mean ± Standard deviation | | (95% CI) | |
| Preoperative | 775.2 ± 137.5 | 837.7 ± 192.4 | −62.50 (−194.66 to 69.66) | |
| 1 month | 571.2 ± 81.2† | 548.0 ± 71.7† | 23.20 (−34.92 to 81.32) | .387 |
| 3 months | 582.8 ± 71.7† | 528.1 ± 70.4† | 54.70 (0.25 to 109.15) | .634 |
| 6 months‡ | 568.7 ± 82.2† | 534.7 ± 81.5† | 34.00 (−28.79 to 96.79) | .823 |
| 12 months§ | 589.5 ± 89.2† | 558.5 ± 78.7† | 31.00 (−32.82 to 94.82) | .733 |

DSAEK, Descemet stripping automated endothelial keratoplasty; DMEK, Descemet's membrane endothelial keratoplasty.

* Mann-Whitney test; † significant difference with preoperative value; ‡ 6-month analysis: DSAEK n = 16, DMEK n = 10; § 12-month analysis: DSAEK n = 14, DMEK n = 10

**Table 6. Comparison of postoperative complications between DSAEK and DMEK.**

| | Total | DSAEK (n = 16) | DMEK (n = 11) | P value* |
|---|---|---|---|---|
| | n (%) | | | |
| Valve tip revision or reposition† | 2 (7.4%) | 0 (0%) | 2 (18.2%) | 1.000 |
| Air injection | 7 (26%) | 4 (25%) | 3 (27%) | 1.000 |
| Uncontrolled IOP | 6 (22%) | 2 (13%) | 4 (36%) | .187 |
| CME | 2 (7%) | 1 (6%) | 1 (9%) | 1.000† |
| CMV | 4 (15%) | 3 (19%) | 1 (9%) | .624 |

DSAEK, Descemet stripping automated endothelial keratoplasty; DMEK, Descemet's membrane endothelial keratoplasty; IOP: Intraocular pressure; CME: Central macular edema.

* Fisher exact test. † Intraoperatively.

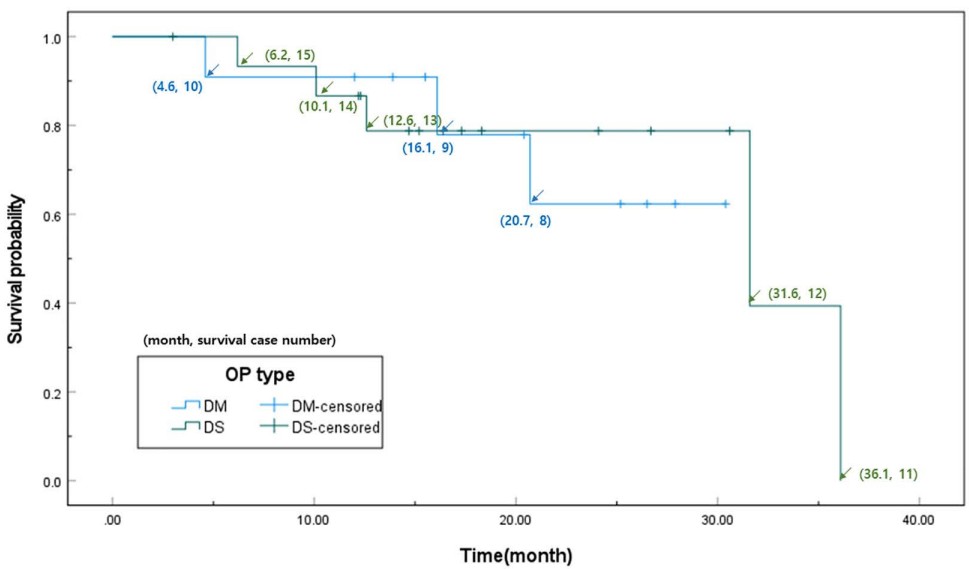

**Fig 1. Kaplan-Meier survival curves comparing graft survival in Descemet stripping automated endothelial keratoplasty (DSAEK) and Descemet membrane endothelial keratoplasty (DMEK).**

The comparable visual outcomes between DSAEK and DMEK in our study contrast with previous reports in uncomplicated cases, where DMEK typically demonstrates superior visual recovery [13,14]. Studies suggest that DMEK's thinner graft profile and more physiological anatomy contribute to better visual outcomes than DSAEK [5]. However, in eyes with GDDs in the present study, several factors may partially obscure the optical advantages of DMEK over DSAEK [8]. Firstly, eyes requiring GDDs often have intrinsically limited visual potential due to advanced glaucoma and chronic ocular comorbidities. Secondly, preoperative central corneal thickness was numerically greater in the DMEK group than in the DSAEK group, suggesting more advanced corneal edema at baseline. Prolonged or severe corneal edema may induce stromal fibrosis, which can constrain postoperative visual recovery regardless of the endothelial keratoplasty technique. Additionally, the DSAEK procedures in our study were performed using ultrathin grafts (mean thickness: 58.14 ± 11.88 μm, range: 45–88 μm), which differ substantially from conventional DSAEK. Previous studies demonstrated that ultrathin DSAEK achieved better visual outcomes than conventional DSAEK and, in selected clinical settings, may approach the visual outcomes of DMEK [13,14]. Taken together, these considerations may help explain the comparable visual outcomes observed in our study and should not be interpreted as evidence of equivalence between DSAEK and DMEK in uncomplicated eyes.

Although unadjusted analysis showed significantly higher postoperative ECD in the DSAEK group at 6 and 12 months (P = 0.040 and P = 0.043, respectively), this difference was no longer significant after covariate adjustment for donor ECD (P = 0.136 and P = 0.149, respectively), suggesting that baseline donor characteristics rather than surgical technique primarily accounted for the observed differences. Notably, donor age was significantly younger in the DSAEK group (52.94 ± 11.17 years vs. 62.36 ± 10.02 years, P = 0.034), with DSAEK donors ranging from 27 to 66 years compared to 55–88 years in the DMEK group. This age distribution reflects differing technical requirements between the procedures. For DMEK, donors aged 55–70 years are generally preferred in clinical practice, as younger donor tissue tends to scroll more tightly due to greater Descemet membrane elasticity, which may increase technical difficulty during graft preparation and anterior chamber unfolding and lead to increased endothelial cell loss, potentially affecting graft survival [15]. In contrast, DSAEK can accommodate a wider donor age range, including younger tissue, as the stromal component provides structural support facilitating handling. Consequently, the younger donor age in the DSAEK group was associated with numerically higher baseline donor ECD (2874.6 ± 200.6 vs. 2715.3 ± 272.6 cells/mm², P = 0.103), which likely explains the initially observed postoperative ECD differences that resolved after statistical adjustment (Table 3). Previous studies have demonstrated associations between younger donor age and superior endothelial cell survival following corneal transplantation [16,17]. However, this association may not directly apply to DMEK, in which the use of older donor tissue may reduce graft manipulation and ECL during procedures. In this context, recent evidence suggests that donor age may serve as a surrogate marker, with intrinsic biological characteristics of donor endothelial cells, including cellular senescence and mitochondrial function, representing the true determinants of postoperative ECD and long-term graft outcomes [18]. Future studies controlling for multiple donor-related confounding factors are warranted to comprehensively elucidate which donor characteristics most critically influence graft survival and longevity in eyes with glaucoma drainage devices.

Regarding postoperative complications, we found no significant differences in IOP elevation requiring additional medications or CME between the two groups. These findings are consistent with previous reports by Alshaker et al. [9] and Kang, *et al.* [19], who also found comparable complication rates between DSAEK and DMEK in eyes with prior glaucoma surgery. The rebubbling rates in our study (25% and 27% in DSAEK and DMEK, respectively) were within the range reported in the literature for eyes with GDDs (15%–33%) [20,21]. The interpretation of graft detachment and rebubbling rates is challenging, as these outcomes are influenced by multiple factors including surgical technique, anterior chamber anatomy, and the presence of GDDs. However, the presence of a GDD has been suggested to increase the risk of graft detachment due to altered aqueous flow dynamics and potential interference with air tamponade [22,23].

Graft failures observed in our study occurred in 5 of the 16 eyes in the DSAEK group and in 3 of the 11 eyes in the DMEK group, which were all secondary failures. Our Kaplan-Meier analysis revealed 1-year survival rates of 87% for DSAEK and 91% for DMEK and 2-year survival rates of 78% for DSAEK and 74% for DMEK. These rates are comparable

 

to those of previous reports of EK in eyes with GDDs, where the 2-year survival rates ranged from 66% to 74% for DSAEK [9,12]. Similar survival rates have been reported in eyes with various types of glaucoma surgery, including DSAEK (71–76%) and DMEK (60–82%) [4,17,24].

However, these rates are notably lower than those reported in uncomplicated cases, where past studies have shown 2-year survival rates of 90–96% for both techniques, highlighting the challenging nature of EK in eyes with prior glaucoma surgeries. Maier *et al.* [12] reported that the presence of GDD was an independent risk factor for graft failure after DMEK, with a hazard ratio of 2.8. Although the presence of GDDs presents unique challenges in EK, the mechanisms that increase the risk of graft failure in these eyes remain unclear. This is thought to be due to mechanical endothelial damage, altered anterior chamber dynamics, and increased inflammatory responses [1,2,25,26].

Our findings suggest comparable visual outcomes, complication rates, and graft survival between the two techniques in this challenging population, though these results should be interpreted cautiously given our limited sample size. While our point estimates suggest comparable outcomes, we cannot definitively exclude the possibility of clinically meaningful differences that would be detectable in a larger long-term cohort. The choice between the two techniques may be based on surgeon's experience, donor tissue availability, and specific anatomical consideration, pending confirmation in larger studies [11,27].

This study has several limitations. First, its small sample size may have limited the generalizability of our findings. Due to limited eligible cases, a two-sided $\alpha = 0.05$ test has 80% power to detect only very large standardized differences (Cohen's $d \approx 1.15$) (calculated by G*power). Therefore, the lack of significant differences should be interpreted cautiously, as potential confounders remain. In this context, the lack of a significant difference in postoperative CCT, even with the inherent stromal thickness of DSAEK grafts, highlights the potential for false-negative findings in a small cohort. Second, the non-randomized design limits causal inference. However, our study reduced the risk of selection bias, although potential differences in patient populations referred to different surgeons cannot be excluded. In this study, each surgeon consistently performed only their preferred technique (DMEK by one surgeon, DSAEK by two surgeons) for this population. Third, certain clinically relevant factors such as peripheral anterior synechiae, iris defects, and other anterior chamber comorbidities could not be assessed. These factors may influence surgical outcomes, particularly in complex cases with prior glaucoma surgery [28]. Finally, our follow-up period may not have captured late complications or failures. Despite these limitations, our study may be useful for keratoplasty surgeons managing eyes with prior AGV implantation. This represents the first direct comparison of DSAEK and DMEK specifically in eyes with AGVs.

In conclusion, our results from limited eligible cases suggest that DSAEK and DMEK may achieve comparable outcomes in eyes with prior AGV implantation, with similar visual improvements, complication rates, and graft survival. These findings suggest that both procedures appear to be viable options for corneal endothelial decompensation in eyes with AGV and that the choice between them can be based on surgeon preference, experience, and donor tissue availability. From a patient perspective, these findings provide preliminary reassurance that both surgical approaches can offer meaningful visual rehabilitation in this complex clinical scenario, though outcomes may be influenced by factors beyond surgical technique alone. Future studies with larger cohorts and longer follow-up periods are essential to confirm these findings and more precisely quantify the relative benefits and risks of each technique in this challenging population.

## Acknowledgments

D-.H.A contributed to the acquisition and analysis of data and the writing of the manuscript. Y-.S.B. contributed to the design and implementation of the research, analysis of the results, and writing of the manuscript. D-.H.A., G.Y., J-.Y.S., and J.U.B. performed the analyses, drafted the manuscript, and designed the figures. S-.H.C. and H-.S.K. helped to interpret the results and worked on the manuscript. All of the authors discussed the results and reviewed the final version of the manuscript.

Artificial intelligence-assisted technology (Large Language Model, Claude) was used to translate the manuscript into English and format it in the journal's style before sending it to an English proofreading company [Editage, www.editage.co.k)].

## Author contributions

**Conceptualization:** Yong-Soo Byun.

**Data curation:** Gyeongmin Yoo, Ji-Yun Song, Jin-Uk Beak, So-Hyang Chung, Hyun-Seung Kim, Yong-Soo Byun.

**Formal analysis:** Yong-Soo Byun.

**Funding acquisition:** Yong-Soo Byun.

**Investigation:** Do-Hyeon An, Yong-Soo Byun.

**Methodology:** Do-Hyeon An.

**Project administration:** Do-Hyeon An.

**Software:** Do-Hyeon An, Yong-Soo Byun.

**Visualization:** Do-Hyeon An, Yong-Soo Byun.

**Writing – original draft:** Do-Hyeon An.

**Writing – review & editing:** Yong-Soo Byun.

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
