## [Decision Letter · Decision Letter 0]

24 Jul 2025

Dear Dr. Byun,

Thank you for submitting your manuscript to PLOS ONE. After careful consideration, we feel that it has merit but does not fully meet PLOS ONE’s publication criteria as it currently stands. Therefore, we invite you to submit a revised version of the manuscript that addresses the points raised during the review process.

We look forward to receiving your revised manuscript.

Kind regards,

Hidenaga Kobashi, M.D., Ph.D.

Academic Editor

PLOS ONE

Journal Requirements:

“This research was partially supported by Basic Science Research Program through the National Research Foundation of Korea(NRF) funded by the Ministry of Education (2022R1C1C1011531).”

3. In the online submission form, you indicated that “All relevant data are within the manuscript and its Supporting Information files. Additional data may be available from the corresponding author upon reasonable request.”

3. We note that your Data Availability Statement is currently as follows: All relevant data are within the manuscript and its Supporting Information files. Additional data may be available from the corresponding author upon reasonable request.

**Additional Editor Comments:**

Your research is very meaningful. The PlosOne reviewers have requested that you make some revisions to your paper before it can be submitted. I agree with that. Let's start with the revision process.

Reviewers' comments:

Reviewer's Responses to Questions

**Comments to the Author**

1. Is the manuscript technically sound, and do the data support the conclusions?

Reviewer #1: Partly

Reviewer #2: Yes

2. Has the statistical analysis been performed appropriately and rigorously?

Reviewer #1: No

Reviewer #2: Yes

3. Have the authors made all data underlying the findings in their manuscript fully available?

Reviewer #1: Yes

Reviewer #2: Yes

4. Is the manuscript presented in an intelligible fashion and written in standard English?

Reviewer #1: Yes

Reviewer #2: Yes

Reviewer #1: Authors described a comparative study in terms of clinical outcomes after two types of endothelial keratoplasty following AGV. Despite the interesting topic, there are some concerns to be raised.

1) Authors should show the statistic power using Gpower to justify the number they used.

2) Although they criticize previous studies due to heterogeneity of etiologies, the current study include various glaucoma. Especially, the number of PEX looks different between two groups. Due to small numbers, there is no significant difference.

3) Authors should add the presence of PAS(peripheral anterior synechiae) in detail in all cases. The existence of PAS might affect the survival rates.

4) Authors should add the lens status in detail in pseudophakic cases. How many scleral fixated IOLs, or sulcus fixated IOLs were included?

5) There is no information about the axial length, or ACD(anterior chamber depth).

6) Authors should describe the location of implanted Ahmed tube in all cases. How many tubes behind iris, anterior chamber, and pars plana?

7) Authors should describe the special technique during endothelial keratoplasty in terms of trimming the tips of tubes in the AC, or not.

8) Discussion

Authors seem to conclude that younger age is only a predictor of graft survival.

Please read the paper, and do consider the discussion again.

Kitazawa K, Toda M, Ueno M, Uehara A, Sotozono C, Kinoshita S. The Biologic

Character of Donor Corneal Endothelial Cells Influences Endothelial Cell Density

Post Successful Corneal Transplantation. Ophthalmol Sci. 2022 Oct

28;3(2):100239. doi: 10.1016/j.xops.2022.100239. PMID: 36846106; PMCID:

PMC9944567.

Reviewer #2: This is a well written retrospective study comparing DSAEK and DMEK in eyes with prior AGV implantation. This is a clinically valuable approach, and addressing the points below will further strengthen its impact.

1. Please clarify how the surgical approach (DSAEK vs. DMEK) was selected. Since DMEK was performed by a single surgeon and DSAEK by two others, could surgeon experience have influenced outcomes?

2. Donor age and ECD were significantly different between groups. Was donor tissue allocated randomly or preferentially? Please explain how donor assignment was handled and whether any donor-related selection bias is possible.

3. Given that ultrathin DSAEK was used (mean graft thickness ~58 µm), please discuss how this may have influenced the comparability to DMEK outcomes. Prior literature shows thinner DSAEK grafts may yield outcomes closer to DMEK.

4. The reported rebubbling rates were relatively high. Were any specific protocols used for air tamponade or postoperative positioning to account for the presence of AGV? If so, please include these details.

5. Was AGV tube position (e.g., angle, anterior chamber, sulcus) recorded and analyzed? This factor may influence graft detachment and endothelial trauma and could explain some postoperative variability.

6. Were systemic comorbidities such as DM or autoimmune conditions equally distributed between groups? These factors may affect corneal healing and graft survival and should be addressed if data are available.

7. Please confirm whether all eligible cases during the study period were included and how selection bias was minimized. Was data extraction blinded to the surgical group or outcome?

8. Given the comparable outcomes, do the authors recommend either technique in resource-limited settings where donor quality or surgical experience may vary?

9. The authors report similar graft survival despite prior AGV placement. How do the authors reconcile this with studies that report lower graft survival in AGV eyes? Please discuss possible protective or confounding factors in this cohort.

**Do you want your identity to be public for this peer review?** For information about this choice, including consent withdrawal, please see our Privacy Policy

Reviewer #1: No

Reviewer #2: **Yes:** Muralidhar Ramappa

---

## [Author Response · Author response to Decision Letter 1]

9 Nov 2025

Manuscript ID: PONE-D-25-33955

Title: Comparative Outcomes of DSAEK and DMEK in Eyes with Prior Ahmed Glaucoma Valve Implantation

Reviewer #1:

Authors described a comparative study in terms of clinical outcomes after two types of endothelial keratoplasty following AGV. Despite the interesting topic, there are some concerns to be raised.

1) Authors should show the statistic power using Gpower to justify the number they used.

- We appreciate your insightful comments that have significantly strengthened our manuscript. We agree this is an important consideration. Due to limited eligible cases, the study recruited 16 and 11 eyes in groups DSAEK and DMEK, respectively. This sample size inherently limits statistical power. With these group sizes, a two-sided α = 0.05 test has 80% power to detect only very large standardized differences (Cohen’s d ≈ 1.15), as calculated by G*Power.

- We added “Mean Difference (95% CI)” values to all comparative tables to present the absolute magnitude and uncertainty of group differences more clearly. We also submitted Hedges’ g values (small sample correction of Cohen’s d) for each parameter to show standardized effect sizes adjusted for small sample bias as a supplement file.

- We have added power analysis to the Limitations section: "Due to limited eligible cases, a two-sided α=0.05 test has 80% power to detect only very large standardized differences (Cohen's d ≈ 1.15) (calculated by G*power). Therefore, the lack of significant differences should be interpreted cautiously."(lines 263-266). Additionally, we have also revised our Results and Discussion to use more cautious language when interpreting our findings.

2) Although they criticize previous studies due to heterogeneity of etiologies, the current study include various glaucoma. Especially, the number of PEX looks different between two groups. Due to small numbers, there is no significant difference.

We acknowledge this concern. While our cohort includes various glaucoma diagnoses, we have several important points, the distribution of glaucoma types was statistically comparable between groups (P=0.545), including PEX (DSAEK: 2 cases [12.5%] vs DMEK: 2 cases [18.2%]). All eyes shared the common feature of AGV-induced endothelial decompensation, which was our primary focus. This is now emphasized in the Discussion (lines 192-201). We believe that our study's strength lies in focusing exclusively on a single type of drainage device (AGV), despite the heterogeneity of etiologies.

3) Authors should add the presence of PAS(peripheral anterior synechiae) in detail in all cases. The existence of PAS might affect the survival rates.

- We completely agree that PAS is a clinically important factor. Unfortunately, due to the retrospective nature of this study, detailed PAS documentation was not consistently available in all medical records. We have now acknowledged this limitation explicitly (lines 270-273): "Third, certain clinically relevant factors such as peripheral anterior synechiae, iris defects, and other anterior chamber comorbidities could not be assessed. These factors may influence surgical outcomes, particularly in complex cases with prior glaucoma surgery." We have also added Schoenberg et al.'s reference (Ref. 27), which demonstrated that high DSAEK failure rates in AGV eyes are associated with 360° PAS.

4) Authors should add the lens status in detail in pseudophakic cases. How many scleral fixated IOLs, or sulcus fixated IOLs were included?

- We thank the reviewer for this helpful suggestion. We have now described the lens status in pseudophakic eyes in detail in Table 1, including the number of in-the-bag IOLand scleral-fixated IOLs (There were no eyes with the sulcus placed IOL).

5) There is no information about the axial length, or ACD(anterior chamber depth).

- We thank the reviewer for this helpful suggestion. Axial length and anterior chamber depth have now been added in Table 1.

6) Authors should describe the location of implanted Ahmed tube in all cases. How many tubes behind iris, anterior chamber, and pars plana?

- We thank the reviewer for this important comment. We have clarified this important surgical detail. All 27 eyes had AGV tubes placed in the anterior chamber. None were placed in the posterior chamber (behind iris) or pars plana. This is now explicitly stated in the Results section (line 123):"All implanted AGV tubes were placed in the anterior chamber."

7) Authors should describe the special technique during endothelial keratoplasty in terms of trimming the tips of tubes in the AC, or not.

- Excellent observation. Two patients underwent Ahmed valve trimming during endothelial keratoplasty, while valve repositioning was not necessary in the other cases. We have added detailed information about valve tip management in both the Results and Complications sections (lines 167-172): "In two DMEK cases, valve tip trimming was required during surgery, because the valve tip could potentially interfere with DMEK graft unfolding procedure. One of these patients subsequently underwent valve tip repositioning into the sulcus due to uncontrolled IOP elevation and iris chafing. Nevertheless, graft survival was favorable, with a ECD >1000 cells/mm² at postoperative month 12, comparable to the overall cohort."

8) Discussion

Authors seem to conclude that younger age is only a predictor of graft survival.

Please read the paper, and do consider the discussion again.

Kitazawa K, Toda M, Ueno M, Uehara A, Sotozono C, Kinoshita S. The Biologic

Character of Donor Corneal Endothelial Cells Influences Endothelial Cell Density

Post Successful Corneal Transplantation. Ophthalmol Sci. 2022 Oct

28;3(2):100239. doi: 10.1016/j.xops.2022.100239. PMID: 36846106; PMCID:

PMC9944567.

- Thank you for this important reference. We now acknowledge that donor age is likely a surrogate for underlying cellular biological properties rather than a direct causal factor. We have carefully reviewed Kitazawa et al.'s paper and substantially revised our Discussion to incorporate these insights (lines 227-232, Ref 17): "However, recent evidence suggests that donor age may serve as a surrogate marker, with intrinsic biological characteristics of donor endothelial cells, including cellular senescence and mitochondrial function, representing the true determinants of postoperative ECD and long-term graft outcomes.[17] Future studies controlling for multiple donor-related confounding factors are warranted to comprehensively elucidate which donor characteristics most critically influence graft survival and longevity in eyes with glaucoma drainage devices."

Reviewer #2:

This is a well written retrospective study comparing DSAEK and DMEK in eyes with prior AGV implantation. This is a clinically valuable and addressing below points will further strengthen its impact.

1. Please clarify how the surgical approach (DSAEK vs. DMEK) was selected. Since DMEK was performed by a single surgeon and DSAEK by two others, could surgeon experience have influenced outcomes?

- We thank the reviewer for this thoughtful comment. We acknowledge that surgeon experience might have influenced the outcomes. However, each surgeon selected their preferred surgical method, which we believe helped minimize bias arising from variability in surgical proficiency. This point has been addressed in the Methods (line 91-93) and Discussion section (line 267-270) as a potential limitation.

2. Donor age and ECD were significantly different between groups. Was donor tissue allocated randomly or preferentially? Please explain how donor assignment was handled and whether any donor-related selection bias is possible.

- We thank the reviewer for raising this important point. The donor age difference was not due to preferential allocation but reflects inherent technical requirements of each procedure. We have extensively explained this in the Discussion (lines 218-223): " This age distribution reflects differing technical requirements between the procedures. For DMEK, donors aged 55-70 years are generally preferred because younger donor tissue exhibits greater Descemet membrane elasticity, causing tighter graft scrolling that complicates both preparation and anterior chamber unfolding. In contrast, DSAEK can accommodate a wider donor age range, including younger tissue, as the stromal component provides structural support facilitating handling. "

3, Given that ultrathin DSAEK was used (mean graft thickness ~58 µm), please discuss how this may have influenced the comparability to DMEK outcomes. Prior literature shows thinner DSAEK grafts may yield outcomes closer to DMEK.

- We agree with the reviewer that the relatively thin grafts used in our UT-DSAEK group (mean thickness ~58 µm) could have influenced comparability with DMEK. Prior literature has consistently shown that ultrathin DSAEK approaches the visual outcomes of DMEK, with thinner grafts associated with better visual acuity and lower higher-order aberrations compared with conventional DSAEK. Therefore, our UT-DSAEK results may partially reflect this convergence toward DMEK outcomes, which should be taken into account when interpreting the findings. We have added substantial discussion of this important factor (lines 206-211): "Additionally, we performed ultrathin DSAEK (mean graft thickness: 58.14 ± 11.88 μm, range: 45–88 μm), which uses a much thinner graft than conventional DSAEK. Previous studies demonstrated that ultrathin DSAEK achieved better visual outcomes than conventional DSAEK, with results comparable to DMEK.[13,14] These factors may explain the comparable visual outcomes between DSAEK and DMEK observed in our study."

4. The reported rebubbling rates were relatively high. Were any specific protocols used for air tamponade or postoperative positioning to account for the presence of AGV? If so, please include these details.

- We did not employ any specific modifications in surgical protocols for air tamponade or postoperative management in eyes with AGV. However, in order to maintain the intracameral air bubble position and to prevent premature escape of air, patients were more strictly instructed to remain in the supine position immediately after surgery. We mentioned this in Methods section. The rebubbling rates (25% DSAEK, 27% DMEK) are within the reported range for eyes with GDDs (15-33%), as we discuss in lines 236-241.

5. Was AGV tube position (e.g., angle, anterior chamber, sulcus) recorded and analyzed? This factor may influence graft detachment and endothelial trauma and could explain some postoperative variability.

- We thank the reviewer for this important comment. In all cases, the AGV tube was in the anterior chamber. This is now explicitly stated in the Results section (line 123):"All implanted AGV tubes were placed in the anterior chamber."

6. Were systemic comorbidities such as DM or autoimmune conditions equally distributed between groups? These factors may affect corneal healing and graft survival and should be addressed if data are available.

- We thank the reviewer for this important comment. We have added systemic comorbidities that could be listed in the chart to Table 1: Diabetes mellitus: DSAEK 3 (18.8%), DMEK 2 (18.2%), Hypertension: DSAEK 6 (37.5%), DMEK 3 (27.3%), Autoimmune disease: DSAEK 1 (6.3%), DMEK 0 (0%); There were no statistically significant differences between the two groups (P=1.000).

7. Please confirm whether all eligible cases during the study period were included and how selection bias was minimized. Was data extraction blinded to the surgical group or outcome?

- We thank the reviewer for this important comment. All eligible cases with well-documented follow-up records for at least 12 months after surgery were included in the analysis. All consecutive AGV cases meeting inclusion criteria during the study period were included, with no exclusions based on surgical outcomes. To minimize selection bias, data extraction was performed in a blinded manner with respect to both surgical group and outcomes. We have added this important methodological detail to Methods (lines 88-90): "Data extraction and analysis were performed by independent observers who were not involved in surgical procedures or clinical care to minimize selection bias."

8. Given the comparable outcomes, do the authors recommend either technique in resource-limited settings where donor quality or surgical experience may vary?

- We thank the reviewer for this helpful suggestion. Considering that our study demonstrated comparable outcomes, we agree that surgical and resource factors should guide the choice in resource-limited settings. In resource-limited settings, DSAEK may offer advantages due to: wider donor age acceptability (our DSAEK donors ranged 27-66 years vs DMEK 55-88 years), potentially easier learning curve with more forgiving surgical technique, comparable outcomes when using ultrathin grafts. However, DMEK remains excellent when appropriate donor tissue and surgical expertise are available. Since it was not possible to generalize the experiences and conditions of individual practitioners, we only noted that both are viable options for AGV eyes. We address this in our revised Conclusion (lines 278-280):". These findings suggest that both procedures appear to be viable options for corneal endothelial decompression in eyes with AGV and that the choice between them can be based on surgeon preference, experience, and donor tissue availability. "

9. The authors report similar graft survival despite prior AGV placement. How do the authors reconcile this with studies that report lower graft survival in AGV eyes? Please discuss possible protective or confounding factors in this cohort.

- Our 2-year survival rates (78% for DSAEK, 74% for DMEK) are notably lower than those reported in uncomplicated cases (90-96%), confirming that AGV presence significantly compromises graft survival. Our rates fall within the range reported by other GDD studies (66-82% at 2 years), rather than exceeding them. Importantly, we do not claim that our cohort had unexpectedly good outcomes or protective factors against AGV-related graft failure. Rather, our findings show that: Both techniques demonstrated the expected lower survival in AGV eyes compared to uncomplicated cases. Within this high-risk population, the two techniques performed comparably to each other. Both techniques remain viable options when managing this challenging patient population.

---

## [Decision Letter · Decision Letter 1]

10 Dec 2025

Dear Dr. Byun,

Thank you for submitting your manuscript to PLOS ONE. After careful consideration, we feel that it has merit but does not fully meet PLOS ONE’s publication criteria as it currently stands. Therefore, we invite you to submit a revised version of the manuscript that addresses the points raised during the review process.

We look forward to receiving your revised manuscript.

Kind regards,

Hidenaga Kobashi, M.D., Ph.D.

Academic Editor

PLOS One

Journal Requirements:

**Additional Editor Comments:**

Your paper is moving forward positively towards acceptance, but there are a few points that need to be pointed out. We have carefully read the reviewer's comments and look forward to your response.

Reviewers' comments:

Reviewer's Responses to Questions

**Comments to the Author**

Reviewer #1: All comments have been addressed

Reviewer #3: (No Response)

2. Is the manuscript technically sound, and do the data support the conclusions?

Reviewer #1: Yes

Reviewer #3: Partly

3. Has the statistical analysis been performed appropriately and rigorously?

Reviewer #1: Yes

Reviewer #3: Yes

4. Have the authors made all data underlying the findings in their manuscript fully available?

Reviewer #1: Yes

Reviewer #3: No

5. Is the manuscript presented in an intelligible fashion and written in standard English?

Reviewer #1: Yes

Reviewer #3: No

Reviewer #1: The author respond to all concerns properly. The manuscript has been much improved. Despite the small number of cases included, it is valuable to show such an interesting clinical comparison.

Reviewer #3: Peer review of “Comparative Outcomes of DSAEK and DMEK in Eyes with Prior Ahmed Glaucoma Valve 2 Implantation”

This study describes the visual and graft outcomes of DSAEK and DMEK in eyes that bear an Ahmed Glaucoma Valve. Despite its small sample size, it adds to the literature, which is limited to a few studies on DMEK alone or DSAEK alone in glaucoma drainage device (GDD) cases or studies that compare the two techniques but do not distinguish between trabeculotomy and GDD. The paper is generally well-written, although some key aspects were not clearly indicated. However, the most important aspects of keratoplasty (visual outcome, endothelial cell loss, perioperative complications, and graft survival) were described.

Major points

1. The DMEK patients had thicker preoperative CCTs than the DSAEK patients (838 vs 775 µm). This suggests that the extent of edema was worse in the DMEK patients. Intense and/or prolonged edema can lead to stromal fibrosis, which can significantly limit visual recovery after DMEK or DSAEK. Thus, the equivalent outcomes of DMEK and DSAEK in the present study may reflect worse bullous keratopathy and stromal fibrosis in the DMEK cases. It is possible that the visual outcomes would have been better in DMEK if disease severity in the two groups was equivalent. This should be discussed.

2. In relation to this, it is suggested in the Discussion that UT-DSAEK can achieve equivalent visual outcomes as DMEK. This is actually a very contested area, with as many studies showing that DMEK outperforms UT-DSAEK (e.g. the RCTs in doi:10.1097/ICO.0000000000002601 and doi:10.1016/j.ophtha.2018.05.019, and the fellow DMEK:DSAEK eye studies in doi:10.1155/2015/750567 and doi:10.1016/j.ophtha.2020.12.021). Lines 206-209 should be phrased more cautiously.

3. The Methods should explictly indicate how glaucoma diagnosis was made and what categories were included (POAG, etc).

4. The Methods should explictly state that cases with 1 or ≥2 AGV and mixed AGV:trabeculectomy/MIGs were included.

5. The Methods should make clear that the patients were “all consecutive patients” and that no DMEK/DSAEK eyes with AGV were excluded for any reason.

6. The postoperative ECD data should also be expressed as %ECL. It is a commonly used measure that accounts for preoperative ECD.

7. The fact that the statistical comparisons in Table 4 did not detect the thicker CCT in DSAEK (due to the extra stroma in the graft) demonstrates the risk of false negatives with small sample sizes. In a larger cohort, this difference would be statistically significant. This point should be included in the Limitations section.

8. Line 167 “In two DMEK cases, valve tip trimming was required during surgery, because the valve tip could potentially interfere with DMEK graft unfolding procedure”.

How was it determined that the valve tip could interfere with DMEK unfolding? Was it after the DMEK graft was introduced into the AC? Or can such interference be detected before the procedure? If the latter, DSAEK may be preferable?

9. There was patient attrition due to graft failure (5 DSAEK and 3 DMEK eyes). When these graft failures occurred during the 12-month study period should be shown with a patient distribution figure. This attrition will also have affected the eye numbers in Tables 2-4. How many eyes were left at the various timepoints should be indicated in these tables.

10. How DSAEK and DMEK in the study compare to uncomplicated DSAEK/DMEK should be briefly described in the Discussion. Does the presence of AGV drastically hamper visual recovery, induce higher ECL, and reduce graft survival?

11. A brief description of how the study findings compare to previous studies (refs 8-12) is needed. A table summarizing the key features of these studies can help with this.

Minor points:

1. Line 73 “This retrospective study analyzed patients who visited at Seoul St.Mary’s Hospital 28/07/2022 and 27/07/2023 and underwent EK at Seoul St. Mary's Hospital between 01/01/2020 and 31/12/2022.”

The relevance of the first-cited but later visit (between 28/07/2022 and 27/07/2023) is unclear. Only the period when the actual surgery was conducted is relevant. The follow-up timepoints (eg. Day 1, Month 1 etc) should also be explictly stated in the Methods.

2. Line 81 “Medical records were reviewed for slit-lamp findings, best-corrected visual acuity (BCVA), intraocular pressure (IOP), surgical history, ocular and systemic comorbidities, ECD, CCT, and postoperative complications, including rebubbling, IOP elevation, and cystoid macular edema (CME).”

When postoperative BCVA, ECD, CCT, IOP, graft attachment, and the macula were examined should be indicated here. The instruments used for these measurements and their manufacturers should also be indicated. How AXL and ACD were measured should also be indicated.

3. Line 86 “Secondary graft failure was defined as irreversible corneal edema following initial improvement.”

This should be defined more precisely. Eg “Secondary graft failure was defined as the reemergence of corneal edema, after initial improvement, that arose at any time point during the 12-month study period.”

4. Line 87 “Postoperative IOP elevation was assessed based on the need for additional glaucoma medications and an IOP exceeding 21 mm Hg.”

Does this relate to only the first week? The postoperative period relating to this should be specified.

5. Line 93 “Any specific modifications were employed.”

The meaning of this was unclear. Please clarify.

6. Line 93 “For 94 DSAEK, precut donor tissue with a graft thickness less than 100 µm was obtained from a nonprofit eye bank” & Line 104 “For DMEK, precut and preloaded grafts in bent-pipette glass tubes were obtained from a nonprofit eye 105 bank”

How were the grafts stored and transported? Which medium, which temperature?

7. Line 96 “Following viscoelastic injection, Descemet's membrane was scored along the circular mark and carefully stripped using a modified reverse Sinskey 98 hook. Following viscoelastic injection, Descemet's 97 membrane was scored along the circular mark and carefully stripped using a modified reverse Sinskey 98 hook.”

The manufacturer details of the viscoelastic and hook should be provided.

8. Line 107 “Filtered room air was used for graft attachment, followed by 2 to 4 hours of supine positioning. Patients were strictly instructed to remain in the supine position and avoid eye rubbing after surgery. Rebubbling with filtered room air or 20% sulfur hexafluoride was performed when at least one-third of the graft was found to be detached postoperatively”

Why was SF6 used for rebubbling but not the endotamponade? It lasts a lot longer than air, generally one would use it for endotamponade, not for rebubbling grafts that are already partially attached.

9. Were the 3 phakic cases left phakic, or was the triple procedure conducted? This should be specified in the Methods.

10. The degree of graft detachment that was considered to indicate the need for rebubbling should be defined. Generally 30% area detached and/or central detachment are considered indications for rebubbling.

11. Line 169 “One of these patients subsequently underwent valve tip repositioning into the sulcus due to 170 uncontrolled IOP elevation and iris chafing.”

When during the 12-month follow up did this occur?

12. Were the uncontrolled IOP episodes all controlled eventually?

13. Line 198 “all eyes shared the common feature of AGV-induced endothelial decompensation”.

This should be phrased more cautiously, eg “all eyes exhibited endothelial decompensation, likely due to the AGV”

14. Line 202 “...contrast with previous 203 reports in uncomplicated cases, where DMEK typically demonstrates superior visual recover”.

References are needed for this statement.

15. Line 219 “For DMEK, donors aged 55-70 years are generally preferred because younger donor tissue exhibits greater Descemet membrane elasticity, causing tighter graft scrolling that complicates both preparation and anterior chamber unfolding.”

References are needed for this statement.

16. Line 219 “For DMEK, donors aged 55-70 years are generally preferred because younger donor tissue 220 exhibits greater Descemet membrane elasticity, causing tighter graft scrolling that complicates both preparation and anterior chamber unfolding.”

It should be indicated here that these complications induce ECL, which in turn promotes graft failure.

17. Line 226 “Previous studies have demonstrated associations between younger donor age and superior endothelial cell survival following corneal transplantation15,16.”

Refs 15 and 16 relate to DSAEK and PKP. It is likely that selecting older donors for DMEK – which leads to less graft handling and ECL - counterbalances the effect of older donor age. This should be indicated here.

18. Line 238 “The graft detachment/rebubbling rate is difficult to conclude”

This is unclear and should be clarified.

19. Line 267 “However, our study reduced the risk of selection bias,”

Briefly summarize here how your study design reduced the risk of selection bias.

20. Line 274 “Despite these limitations, our study provides meaningful contributions to clinical practice.”

This should be more cautious eg “our study may be useful for keratoplasty surgeons dealing with eyes bearing an AGV.”

Typos

1. Line 67: “This study aimed to address this knowledge gap by comparing the 68 outcomes of DSAEK and DMEK in eyes with single type of GDD devices, or AGVs.”

This should read “...with a single type of GDD, namely, AGVs”

1. Line 112 “Mann-Whiteny U test”

1. The AGV ≥2 data are not on the same row as the row title in Table 1.

2. Line 132 “All values are presented as mean ± standard deviation. percentage (%).”

3. Line 194 “we specifically focused on eyes with a single type of glaucoma drainage device, or AGV,”

This should read “eyes with a single type of GDD, namely, AGV,”

4. Line 265 “Therefore, the lack of significant differences should be interpreted cautiously. s potential confounders.

6. Line 270 “population Third,”

7. Line 278 “These findings suggest that both procedures appear to be viable options for corneal endothelial decompression in eyes with AGV”

Decompression should be decompensation.

8. Quotation mark at Line 285.

**Do you want your identity to be public for this peer review?** For information about this choice, including consent withdrawal, please see our Privacy Policy

Reviewer #1: No

Reviewer #3: No

---

## [Author Response · Author response to Decision Letter 2]

17 Jan 2026

Response to Reviewer

We sincerely thank the reviewer for the careful re-evaluation of our revised manuscript and for the insightful and constructive comments. We have carefully addressed all points raised and believe that the manuscript has been substantially strengthened as a result. Below, we provide a point-by-point response.

Major Points

1.The DMEK patients had thicker preoperative CCTs than the DSAEK patients (838 vs 775 µm). This suggests that the extent of edema was worse in the DMEK patients. Intense and/or prolonged edema can lead to stromal fibrosis, which can significantly limit visual recovery after DMEK or DSAEK. Thus, the equivalent outcomes of DMEK and DSAEK in the present study may reflect worse bullous keratopathy and stromal fibrosis in the DMEK cases. It is possible that the visual outcomes would have been better in DMEK if disease severity in the two groups was equivalent. This should be discussed.

Response:

We agree with the reviewer that the greater preoperative CCT in the DMEK group may reflect more advanced corneal edema and potentially greater stromal fibrosis, which could limit postoperative visual recovery. We have now explicitly addressed this important point in the Discussion and acknowledged baseline disease severity as a potential confounder that may have attenuated the visual advantage of DMEK.

Manuscript change (Discussion): “Secondly, preoperative central corneal thickness was numerically greater in the DMEK group than in the DSAEK group, suggesting more advanced corneal edema at baseline. Prolonged or severe corneal edema may induce stromal fibrosis, which can constrain postoperative visual recovery regardless of the endothelial keratoplasty technique.”

2. In relation to this, it is suggested in the Discussion that UT-DSAEK can achieve equivalent visual outcomes as DMEK. This is actually a very contested area, with as many studies showing that DMEK outperforms UT-DSAEK (e.g. the RCTs in doi:10.1097/ICO.0000000000002601 and doi:10.1016/j.ophtha.2018.05.019, and the fellow DMEK:DSAEK eye studies in doi:10.1155/2015/750567 and doi:10.1016/j.ophtha.2020.12.021). Lines 206-209 should be phrased more cautiously.

Response:

We appreciate this important clarification. We have revised the relevant sentences to adopt a more cautious tone and to clearly distinguish our findings in AGV-bearing eyes from evidence derived from uncomplicated cases and randomized controlled trials. We now emphasize that our findings should not be interpreted as equivalence in general, but rather as comparable outcomes in a highly specific and complex clinical context.

Manuscript change (Discussion):

“Taken together, these considerations may help explain the comparable visual outcomes observed in our study and should not be interpreted as evidence of equivalence between DSAEK and DMEK in uncomplicated eyes.”

3. The Methods should explictly indicate how glaucoma diagnosis was made and what categories were included (POAG, etc).

Response:

We agree and have clarified the diagnostic criteria and glaucoma subtypes included in the Methods section.

Manuscript change (Methods): “Glaucoma diagnosis was based on prior clinical diagnosis by glaucoma specialists and included primary open-angle glaucoma, secondary glaucoma, uveitic glaucoma, pseudoexfoliative glaucoma, neovascular glaucoma, and iridocorneal endothelial syndrome.”

4. The Methods should explictly state that cases with 1 or ≥2 AGV and mixed AGV:trabeculectomy/MIGs were included.

Response:

We have revised the Methods to explicitly state that eyes with single or multiple AGVs, as well as those with additional prior glaucoma surgeries, were included.

Manuscript change (Methods): “Eyes with a single AGV, multiple AGVs (≥2), and those with prior trabeculectomy or microinvasive glaucoma surgery in addition to AGV implantation were included.”

5. The Methods should make clear that the patients were “All eligible cases were consecutively included, with no exclusions among eyes with prior AGV implantation undergoing DSAEK or DMEK.”

Response:

We agree and have explicitly stated this in the Methods.

Manuscript change (Methods): “All eligible cases during the study period were consecutively included, and no eyes with prior AGV implantation undergoing DSAEK or DMEK were excluded for any reason.”

6. The postoperative ECD data should also be expressed as %ECL. It is a commonly used measure that accounts for preoperative ECD.

Response:

We agree with the reviewer and have additionally calculated postoperative endothelial cell loss as percentage endothelial cell loss (%ECL), using donor endothelial cell density as the reference. This measure was included to account for baseline donor ECD and to allow a complementary assessment of endothelial outcomes. A new table (Table 4) has been added to present %ECL values at each postoperative time point. As shown, %ECL did not differ significantly between the DSAEK and DMEK groups at any time point. Corresponding descriptions have been added to both the Methods and Results sections.

Manuscript changes:

Methods: “Postoperative endothelial cell loss (ECL) was additionally calculated and expressed as a percentage (%ECL) relative to the donor ECD using the following formula: %ECL = [(donor ECD − postoperative ECD) / donor ECD] × 100.”

Results: “Postoperative %ECL showed no statistically significant differences between the DSAEK and DMEK groups at 1, 3, 6, or 12 months postoperatively (Table 4).” A new table (Table 4) summarizing %ECL comparisons between DSAEK and DMEK was added.

7. The fact that the statistical comparisons in Table 4 did not detect the thicker CCT in DSAEK (due to the extra stroma in the graft) demonstrates the risk of false negatives with small sample sizes. In a larger cohort, this difference would be statistically significant. This point should be included in the Limitations section.

Response:

We agree with the reviewer and have explicitly addressed this point in the Limitations section.

Manuscript change (Limitations): “In this context, the lack of a significant difference in postoperative CCT, even with the inherent stromal thickness of DSAEK grafts, highlights the potential for false-negative findings in a small cohort.”

8. Line 167 “In two DMEK cases, valve tip trimming was required during surgery, because the valve tip could potentially interfere with DMEK graft unfolding procedure”. How was it determined that the valve tip could interfere with DMEK unfolding? Was it after the DMEK graft was introduced into the AC? Or can such interference be detected before the procedure? If the latter, DSAEK may be preferable?

Response:

We appreciate this request for clarification. Valve tube length was assessed intraoperatively before graft insertion. When the AGV tube was excessively long on intraoperative assessment, valve tip trimming was performed before graft insertion to avoid interference with graft positioning. This decision was based on the surgeon’s anatomical assessment rather than the choice of endothelial keratoplasty technique.

Manuscript change (Methods): “When the AGV tube was excessively long on intraoperative assessment, valve tip trimming was performed before graft insertion to avoid interference with graft positioning.”

9. There was patient attrition due to graft failure (5 DSAEK and 3 DMEK eyes). When these graft failures occurred during the 12-month study period should be shown with a patient distribution figure. This attrition will also have affected the eye numbers in Tables 2-4. How many eyes were left at the various timepoints should be indicated in these tables.

Response:

We thank the reviewer for this important comment and have clarified these points in the revised manuscript. The Kaplan–Meier survival curve (Figure 1) includes all graft failure events observed over the entire follow-up period, and the timing of all failures (5 DSAEK and 3 DMEK eyes) is now explicitly indicated in the figure, together with the number of remaining eyes at each time point.

The outcome analyses presented in Tables 2 – 5 (newly numbered tables) were designed from the outset to represent a 12-month cohort. At each postoperative visit, eyes that had experienced graft failure by that time point were excluded from the analysis. Accordingly, the number of eyes analyzed differed across time points and is now explicitly reported in the Results and in table footnotes (DSAEK/DMEK: 16/11 at 1 week, 1 month, and 3 months; 16/10 at 6 months; and 14/10 at 12 months).

10. How DSAEK and DMEK in the study compare to uncomplicated DSAEK/DMEK should be briefly described in the Discussion. Does the presence of AGV drastically hamper visual recovery, induce higher ECL, and reduce graft survival?

Response:

This comparison is briefly addressed in the Discussion. We noted that, unlike uncomplicated endothelial keratoplasty where DMEK typically demonstrates superior visual outcomes and higher graft survival, eyes with prior AGV implantation showed attenuated visual recovery and reduced graft survival regardless of the EK technique. These findings emphasize that the presence of an AGV represents a challenging anterior chamber environment compared with uncomplicated cases.

11. A brief description of how the study findings compare to previous studies (refs 8-12) is needed. A table summarizing the key features of these studies can help with this.

Response:

We appreciate the reviewer’s suggestion. Prior studies (refs 8–12) are already cited and discussed in the Discussion to contextualize our findings. We chose not to add a summary table because the referenced studies are highly heterogeneous, and presenting only selected studies in tabular form could introduce selection bias and imply a level of systematic review beyond the scope of this original research article.

Minor points

1. Line 73 “This retrospective study analyzed patients who visited at Seoul St.Mary’s Hospital 28/07/2022 and 27/07/2023 and underwent EK at Seoul St. Mary's Hospital between 01/01/2020 and 31/12/2022.” The relevance of the first-cited but later visit (between 28/07/2022 and 27/07/2023) is unclear. Only the period when the actual surgery was conducted is relevant. The follow-up timepoints (eg. Day 1, Month 1 etc) should also be explictly stated in the Methods.

Response:

We agree with the reviewer. The visit period was unclear and potentially confusing. We have removed this information and revised the Methods to specify only the surgical period. We have also explicitly stated the postoperative follow-up time points.

Manuscript change (Methods):

“This retrospective study included patients who underwent endothelial keratoplasty between 01/01/2020 and 31/12/2022, with postoperative evaluations performed at regular follow-up visits during the 12-month postoperative period.”

2. Line 81 “Medical records were reviewed for slit-lamp findings, best-corrected visual acuity (BCVA), intraocular pressure (IOP), surgical history, ocular and systemic comorbidities, ECD, CCT, and postoperative complications, including rebubbling, IOP elevation, and cystoid macular edema (CME).” When postoperative BCVA, ECD, CCT, IOP, graft attachment, and the macula were examined should be indicated here. The instruments used for these measurements and their manufacturers should also be indicated. How AXL and ACD were measured should also be indicated.

Response:

We appreciate this suggestion and have expanded the Methods to specify the timing, measurement devices, and manufacturers.

Manuscript change (Methods):

“Postoperative BCVA, IOP, ECD, and CCT were assessed at every visit throughout the 12-month postoperative period. BCVA was measured using a Snellen chart and converted to logMAR units. IOP was measured using a non-contact pneumotonometer (CT-80A; Canon Inc., Tokyo, Japan). ECD and CCT were assessed using specular microscopy (CellChek XL; Konan Medical, Nishinomiya, Japan) and an ultrasound pachymeter (SP-3000; Tomey Corporation, Nagoya, Japan), respectively. Axial length (AL) and anterior chamber depth (ACD) were measured preoperatively using optical biometry (IOLMaster 700; Carl Zeiss Meditec, Jena, Germany). Graft attachment was evaluated postoperatively using slit-lamp biomicroscopy and anterior segment optical coherence tomography (ANTERION; Heidelberg Engineering, Heidelberg, Germany). Retinal and macular status were evaluated at postoperative 1 and 3 using ultra-widefield imaging (Optos Silverstone; Optos PLC, Dunfermline, UK) and spectral-domain OCT (Heidelberg Retina Angiograph OCT; Heidelberg Engineering, Heidelberg, Germany).”

3. Line 86 “Secondary graft failure was defined as irreversible corneal edema following initial improvement.”

This should be defined more precisely. Eg “Secondary graft failure was defined as the reemergence of corneal edema, after initial improvement, that arose at any time point during the 12-month study period.”

Response:

We agree and have revised the definition to be more precise.

Manuscript change (Methods):

“Secondary graft failure was defined as the reemergence of corneal edema after initial postoperative improvement, occurring at any time during the 12-month follow-up period.”

4. Line 87 “Postoperative IOP elevation was assessed based on the need for additional glaucoma medications and an IOP exceeding 21 mm Hg.”

Does this relate to only the first week? The postoperative period relating to this should be specified.

Response:

Thank you for this clarification request. We have specified that postoperative IOP elevation was assessed throughout the entire follow-up period.

Manuscript change (Methods):

“Postoperative IOP elevation was defined as an IOP exceeding 21 mm Hg and/or the need for additional glaucoma medications at any time during the 12-month postoperative follow-up period.”

5. Line 93 “Any specific modifications were employed.” The meaning of this was unclear. Please clarify.

Response:

We agree that this statement was unclear. As no systematic technique modifications were applied beyond standard approaches, this phrase has been removed.

6. Line 93 “For DSAEK, precut donor tissue with a graft thickness less than 100 µm was obtained from a nonprofit eye bank” & Line 104 “For DMEK, precut and preloaded grafts in bent-pipette glass tubes were obtained from a nonprofit eye bank” How were the grafts stored and transported? Which medium, which temperature?

Response:

We have added details regarding graft storage and transport conditions.

Manuscript change (Methods):

“All donor corneal tissues were preserved in Optisol-GS and transported at 4°C according to standard eye bank protocols.”

7. Line 96 “Following viscoelastic injection, Descemet's membrane was scored along the circular mark and carefully stripped using a modified reverse Sinskey hook. Following viscoelastic injection, Descemet's membrane was scored along the circular mark and carefully stripped using a modified reverse Sinskey hook.” The manufacturer details of the viscoelastic and hook should be provided.

Response:

We have added the manufacturer information for the viscoelastic material (Hyalu®; Hanmi Pharm. Co., Ltd., Seoul, Korea) and a modified reverse Sinskey hook (Moria SA, Antony, France).

8. Line 107 “Filtered room air was used for graft attachment, followed by 2 to 4 hours of supine positioning. Patients were strictly instructed to remain in the supine position and avoid eye rubbing after surgery. Rebubbling with filtered room air or 20% sulfur hexafluoride was performed when at least one-third of the graft was found to be detached postoperatively”

Why was SF6 used for rebubbling but not the endotamponade? It lasts a lot longer than air, generally one would use it for endotamponade, not for rebubbling grafts that are already partially attached.

Response:

Both filtered room air and 20% sulfur hexafluoride are commonly used for initial graft tamponade as well as for rebubbling, depending on surgeon preference and intraoperative or postoperative considerations. In our practice, room air was routinely used for primary graft attachment, whereas either room air or 20% sulfur hexafluoride was selected for rebubbling at the surgeon’s discretion.

9. Were the 3 phakic cases left phakic, or was the triple procedure conducted? This sh

---

## [Decision Letter · Decision Letter 2]

18 Feb 2026

Comparative Outcomes of DSAEK and DMEK in Eyes with Prior Ahmed Glaucoma Valve Implantation

PONE-D-25-33955R2

Dear Dr. Byun,

We’re pleased to inform you that your manuscript has been judged scientifically suitable for publication and will be formally accepted for publication once it meets all outstanding technical requirements.

Kind regards,

Hidenaga Kobashi, M.D., Ph.D.

Academic Editor

PLOS One
---

## [Editor Report · Acceptance letter]

PONE-D-25-33955R2

PLOS One

Dear Dr. Byun,

I'm pleased to inform you that your manuscript has been deemed suitable for publication in PLOS One. Congratulations! Your manuscript is now being handed over to our production team.

Kind regards,

on behalf of

Dr. Hidenaga Kobashi

Academic Editor

PLOS One